# Molecular and Serological Studies on Potential SARS-CoV-2 Infection among 43 Lemurs under Human Care—Evidence for Past Infection in at Least One Individual

**DOI:** 10.3390/ani14010140

**Published:** 2023-12-31

**Authors:** Beatriz Musoles-Cuenca, Jordi Aguiló-Gisbert, Teresa Lorenzo-Bermejo, Rocío Canales, Beatriz Ballester, Umberto Romani-Cremaschi, Rosa Martínez-Valverde, Elisa Maiques, Diana Marteles, Pablo Rueda, Vicente Rubio, Sergio Villanueva-Saz, Consuelo Rubio-Guerri

**Affiliations:** 1Department of Biomedical Sciences, Faculty of Health Sciences, Universidad Cardenal Herrera-CEU, 46113 Valencia, Spain; beatriz.musolescuenca@uchceu.es (B.M.-C.); teresa.lorenzobermejo@uchceu.es (T.L.-B.); beatriz.ballesterllobell@uchceu.es (B.B.); emaiques@uchceu.es (E.M.); 2Servicio de Análisis, Investigación, Gestión de Animales Silvestres (SAIGAS), Veterinary Faculty, Universidad Cardenal Herrera-CEU, 46113 Valencia, Spain; jordi.aguilo@uchceu.es; 3Veterinary Department, Mundomar Benidorm, 03503 Alicante, Spain; veterinario@mundomar.es (R.C.); umberto.romanicremaschi@gmail.com (U.R.-C.); 4Veterinary and Conservation Department, Bioparc Fuengirola, 29640 Fuengirola, Spain; rmartinez@bioparcfuengirola.es; 5Clinical Immunology Laboratory, Veterinary Faculty, University of Zaragoza, 50013 Zaragoza, Spain; diana.martelesaragues@gmail.com (D.M.); ruedazgz_@hotmail.com (P.R.); 6Department of Genomics and Proteomics, Instituto de Biomedicina de Valencia del Consejo Superior de Investigaciones Científicas (IBV-CSIC), 46010 Valencia, Spain; 7Group 739, IBV-CSIC, Centre for Biomedical Network Research, Instituto de Salud Carlos III (CIBERER-ISCIII), 46010 Valencia, Spain; 8Department of Pharmacy, Faculty of Health Sciences, Universidad Cardenal Herrera-CEU, 46113 Valencia, Spain

**Keywords:** COVID-19, SARS-CoV-2, lemurs, captive, *Varecia variegata*, *Lemur catta*, one health, animal COVID-19

## Abstract

**Simple Summary:**

COVID-19, an emerging infectious disease of possible animal origin due to SARS-CoV-2, which has caused a severe pandemic, has also affected some zoo animals. Since non-human primates are considered susceptible hosts, we tested 43 lemurs (20 black-and-white ruffed lemur *Varecia variegata* and 23 ring-tailed lemur *Lemur catta*) from a Spanish zoological institution, which were in close contact with humans during the pandemic period, for SARS-CoV-2 infection in both 2022 and 2023. We used molecular techniques for viral RNA detection in oropharyngeal and rectal swabs and blood anti-SARS-CoV-2 serology. The molecular assays were negative, but one animal was seropositive, strongly suggesting previous infection by SARS-CoV-2 of that animal. These data, while not pinpointing a high susceptibility of lemurs to SARS-CoV-2 infection, add to existing information on the need for surveillance of that virus in animals.

**Abstract:**

In the setting of the recent COVID-19 pandemic, transmission of SARS-CoV-2 to animals has been reported in both domestic and wild animals and is a matter of concern. Given the genetic and functional similarities to humans, non-human primates merit particular attention. In the case of lemurs, generally considered endangered, they are believed to be susceptible to SARS-CoV-2 infection. We have conducted a study for evidence of SARS-CoV-2 infection among the 43 lemurs of Mundomar, a zoological park in Benidorm, Spain. They belong to two endangered lemur species, 23 black-and-white ruffed lemurs (*Varecia variegata*) and 20 ring-tailed lemurs (*Lemur catta*). Health assessments conducted in 2022 and 2023 included molecular analyses for SARS-CoV-2 RNA of oral and rectal swabs using two different RT-qPCR assays, always with negative results for SARS-CoV-2 in all animals. The assessment also included serological testing for antibodies against the receptor-binding domain (RBD) of the spike protein (S) of SARS-CoV-2, which again yielded negative results in all animals except one black-and-white ruffed lemur, supporting prior infection of that animal with SARS-CoV-2. Our data, while not indicating a high susceptibility of lemurs to SARS-CoV-2 infection, show that they can be infected, adding to the existing information body on potential ways for SARS-CoV-2 virus spreading in zoos, highlighting the need for animal surveillance for the virus.

## 1. Introduction

The discovery of the severe acute respiratory syndrome coronavirus 2 (SARS-CoV-2) in Wuhan in late 2019 and its build-up into the global health crisis represented by the COVID-19 pandemic prompted an unparalleled international pursuit encompassing research on the virus, its transmission dynamics, the host’s immune responses, and potential animal reservoirs [1,2]. Multiple studies have substantiated SARS-CoV-2 infection in some domestic and farmed animals, including dogs, cats, mink, and ferrets (see, for example, [3,4,5,6]). The presence of SARS-CoV-2 has been confirmed in some zoo-dwelling animals, including lions, tigers, and great apes [7,8,9]. Sustained intraspecies transmission has been documented among farmed minks [5] and wild white-tailed deer of North America [10]. The virus has also been detected in wild African white rhinoceros [11], feral mink [12], and wild otters [13]. The occurrence of SARS-CoV-2 in wild animals raises concerns about the potential for viral evolution outside the viral vigilance network, with the possibility of re-entry of novel variants in the human population [2]. Furthermore, the occurrence of deadly infections among natural populations of endangered animal species could place these species at increased extinction risk.

Given their genetic closeness to humans, non-human primates were found [8,9,14,15] or predicted [16,17] to be susceptible to SARS-CoV-2 infection. Among the primates deserving particular attention are lemurs, since many lemur species are endangered, and they are known to share infectious diseases with humans and domestic animals when they inhabit tourists-frequented regions and humanized disrupted habitats [18,19,20], and they do not appear particularly resistant to emerging infectious diseases [21]. Furthermore, predictions based on the amino acid sequence of the major SARS-CoV-2 receptor (the ACE2 protein) suggested that lemurs could be highly susceptible to the infection by this virus, particularly lemur species from the *Avahi* and *Propithecus* genera, followed by animals from the *Lemur* and *Varecia* genera [17]. As highlighted by others [17], it appears important to test these predicted susceptibilities. An indirect way of doing so could be by monitoring SARS-CoV-2 infection of lemurs held in zoological facilities in which they frequently come into close contact with zoo visitors and animal keepers. We are not aware of any published prior study on this question on the lemur group of primates. To our knowledge, only one negative serologic test for SARS-CoV-2 in one wild ruffed lemur has been reported thus far [22].

We report here molecular and serological testing for SARS-CoV-2 in 43 lemurs from a zoological institution of a major touristic hub of the Mediterranean seashore of Spain. This park keeps in captivity, although free-roaming, 23 black-and-white ruffed lemurs (*Varecia variegata*) and 20 ring-tailed lemurs (*Lemur catta*). These animals are exposed to very large numbers of human visitors including close contact with visitors for educational reasons. During the COVID-19 pandemic, visits to the park resumed in July 2020 and close contact also resumed in 2021, with the summer of 2022 representing a major visitor peak in the park’s history. In addition, the animals were always in continued contact with their human keepers. 

## 2. Materials and Methods

### 2.1. Animals: Location and Interaction with Humans

Mundomar, a zoological park located in Benidorm (seashore of the Valencian Community, Eastern Spain), hosts approximately 300 animals (marine and terrestrial mammals, birds, and reptiles), among them 43 lemurs of 1–15 years of age, 23 and 20 of them belonging, respectively, to the *V. variegata* and *L. catta* species (two species in the IUCN Red List of Threatened Species as endangered and critically endangered, respectively; https://www.iucnredlist.org (accessed on 18 November 2023)). These lemurs are hosted in a very spacious outdoor enclosure featuring rocks, trunks, branches, hammocks, climbing ropes, and cave shelters. This enclosure is connected with temperature-controlled indoor enclosures that are always available for these animals. The outdoor enclosure is surrounded by a 2.5 m-high glass barrier, with efficient ventilation ensured by roof and top-lateral fences. The animals are in close daily contact with their keepers; although after the emergence of SARS-CoV-2 in 2020, the keepers have had to use personal protective equipment (PPE), including donning filtering facepiece particle 2 (FFP2) masks, plastic face-covering transparent shields, gloves, and boots, and had to disinfect themselves upon entering and exiting the animal enclosures using a water solution of Virkon-S (Zotal Laboratories, Seville, Spain), a bactericidal and viricidal mixture of peroxide compounds, surfactants, and organic acids. The keepers were tested for SARS-CoV-2 upon suspicion of contagion, and since March 2021 they were offered anti-COVID-19 vaccination by the Spanish health system.

Normally, close interaction with lemurs was allowed to visitors for educational purposes (largely feeding and photography opportunities). With the COVID-19 pandemic, the park was closed to visitors for nearly 4 months in 2020 (March 15 to early July). Upon reopening, educational interactions with animals were suspended, with the visitors being required to wear facemasks and to observe the animals from the outside of the enclosure. In 2021, educational interactions resumed in small groups of 20–30 people per day who had to wear facemasks and enter and exit the enclosure through a disinfectant footbath (utilizing Virkon-S), also sanitizing their hands with hydroalcoholic solution before entering the enclosure. By 2022, the biosecurity measures for visitors were simplified to foot and hand sanitization, and the use of facemasks was no longer mandatory, although zookeepers continued to employ PPE when entering primate facilities, including facemasks.

### 2.2. Procurement of Samples

Samples for this study (March 2022 till March 2023) were collected as part of routine health assessments of the animals, performed at least once on all the animals (twice for 18 animals, with 7–11 months spacing between the two assessments, Table 1). Twenty-four animals were assessed after the summer of 2022, when the number of visitors had reached top levels in the parks’ history (around 1000 visitors/day). 

Assessments followed well-established veterinary procedures for animal management and welfare, using gentle manual restraining and light isoflurane anesthesia (delivered via facemask), typically 5 min for weighing (Table 1), recording of body temperature, heart rate, and respiratory frequency, evaluation by inspection of animal’s overall health and potential lesions, and collection of venous blood samples from the jugular vein for hematocrit, hemogram, and standard biochemical determinations in plasma (performed by IDEXX Laboratories, Barcelona, Spain), also obtaining blood serum and preserving it at −20 °C for serological analyses. In addition, oropharyngeal and rectal swabs were collected aseptically during anesthesia and were preserved in closed plastic tubes under Sample Preservation Solution (reference number P042T0020100; from JiangSu Mole Bioscience in Taizhou, China; distributed in Spain by Palex Medical, Madrid), a proprietary solution [12] that inactivates SARS-CoV-2 and preserves RNA for molecular analyses. The sealed tubes were preserved at −80 °C. Table 1 lists the collected samples, noting that there was one animal from each species and two *L. catta* individuals for which serum or oropharyngeal swabs could not be obtained, respectively.

### 2.3. Molecular Analyses

For RNA extraction, we used 0.2 mL of the Sample Preservation Solution that had hosted the nasopharyngeal or rectal swab of each animal and utilized the NZY Total RNA Isolation kit (NZYtech, Lisbon, Portugal). The isolated RNA was stored at −80 °C.

Viral testing was carried out on 5 μL of isolated RNA solution by one-tube RT-qPCR using the commercial Viasure assay (CerTest Biotec, Zaragoza, Spain; distributed by Palex Medical). This test amplifies specific regions of the *ORF1ab* and nucleocapsid (*N*) viral genes and of the host *RNaseP* gene (used as an operational internal positive control). This last gene amplification in this commercial kit was previously shown to work for otters [13], and now we find that it works for lemurs too (see below). 

We also used a second commercial test from another supplier with all isolated RNAs (NZYtech SARS-CoV-2 One-Step RT-PCR kit; from NZYtech, Lisbon, Portugal) for SARS-CoV-2 detection in 8 µL of the isolated RNA solutions. This test targets another viral gene (*RdRp*) in addition to the *N* gene and measures the combined signal from these two viral genes using the same fluorophore for both of them. It also uses the host *RNaseP* gene as an internal control, utilizing a second fluorophore for this gene. Again, preliminary assays proved the suitability of the internal control of this commercial assay for lemur samples, in line with our previous observation that it was appropriate for use in otters [13]. 

We also used with the seropositive animal (animal #6, see below) a highly sensitive home-made two-tube RT-qPCR assay originally reported for targeting the spike protein (*S*) gene in feral mink [12] and a domestic dog [4] and expanded later on to target two additional viral genes (*N* and *ORF10)* [13,23].

We carried out in parallel negative controls as well as positive controls (swab-derived RNA from a positive human) [23]. These controls constantly yielded negative and positive results, respectively.

For all PCR procedures, we used the Aria Mx Real-Time PCR (qPCR) instrument (Agilent Technologies, Santa Clara, CA, USA). All individuals involved in the research that conducted every aspect of the process from sample extraction to detection tested negative for SARS-CoV-2 at the time of the assays.

### 2.4. Serological Studies

An indirect enzyme-linked immunosorbent assay (ELISA) that detects IgG recognizing the receptor-binding domain (RBD) of the spike protein (S) of SARS-CoV-2 virus (ancestral SARS-CoV-2 strain, Wuhan strain) in multiple animal species was used as previously described [24,25]. Unfortunately, despite the prevailing presence of the SARS-CoV-2 Omicron variant (B.1.1.529) during our study period, this specific variant was not accessible in our laboratory for analysis. However, this limitation did not hinder the detection of seropositive ferrets using the same technique within the study period in a previous work [24]. The sera under study were used at 1:100 dilution, utilizing for detection Pierce recombinant protein A/G conjugated to horseradish peroxidase (ref. 32490 from Thermo Fisher Scientific, Waltham, MA, USA) diluted 1:100,000, with colorimetric detection of the peroxidase activity at 492 nm on a Multiskan ELISA reader (Labsystems, Midland, ON, Canada) using ortho-phenylene-diamine as a peroxidase substrate. Each 96-well plate included a positive control, consisting of sera from previously identified seropositive ferret [24] and cat [25], as well as serum from a healthy, non-infected lemur obtained before the COVID-19 pandemic as a negative control.

Before use with the present sera, the technique was optimized using 12 *L. catta* and 4 *V. variegata* sera collected before 2020, prior to the emergence of SARS-CoV-2 (provided by the serum bank of Bioparc-Fuengirola, Fuengirola, Spain), and with sera obtained in 2020 from 9 *L. catta* and 13 *V. variegata* individuals among the study animals, which had tested seronegative for SARS-CoV-2 with confirmation from another laboratory (Veterinary Faculty, University of Cordoba). The cut-off for our present indirect ELISA assay for anti-RBD IgG for lemurs (*L. catta* and *V. variegata*) was set to 0.16 Optical Density units (OD units) (mean +3 standard deviations of values from 38 animals, 17 of them *V. variegata* and 21 of them *L. catta*). Thus, any results surpassing the 0.16 OD threshold were considered positive. The control sera were sourced from the collection of sera within the Laboratory of Clinical Immunology at the Faculty of Veterinary Medicine, University of Zaragoza, Spain.

## 3. Results and Discussion

### 3.1. Normal Status of the Animals including the Seropositive Individual 

The animals, of ages 8 months to 20 years at examination (Table 1 and Figure 1), did not present evidence of disease or significant lesions, as judged by clinical veterinary inspection and the analysis of basic constant and blood analytical parameters. Of interest in the context of COVID-19, none of the animals had hyperthermia (monitored by infrared thermometry). Their weights (Figure 1) were within normal ranges for each species in captivity [26,27], with curves that reflected lower body weights in juvenile individuals, and then some increase with mature age, although for the most aged individuals the weight appeared to decrease somewhat. Similar observations including weight decline at older age have been reported previously for *V. variegata* [26]. When the mean weights of adult males and females (age ≥ 2 years) were compared, no significant differences were found between the two sexes for any of the two species, with mean ± SD (for *n* determinations) for *V. variegata* females, 3.58 ± 0.94 kg (*n* = 13) and for males (*n* = 7), 3.44 ± 0.73 kg; for *L. catta* females (*n* = 9), 2.48 ± 0.50 kg and for males (*n* = 16), 2.78 ± 0.61 kg. In this respect, animal #6, the *V. variegata* female individual that tested seropositive for SARS-CoV-2 in its first assessment at 5 years of age (see below), presented typical weights (3.2 and 3.3 kg) in both assessments, which took place 8 months apart, the second of them when the serological results had returned to the upper part of the normality range (see below), suggesting that the infection of this animal with SARS-CoV-2 virus had not had an important impact on its nutritional state, as it could be the case if the infection had triggered prostration and/or fever.

### 3.2. Negativity of the Molecular Analyses for SARS-CoV-2

The RNA samples obtained from oropharyngeal and rectal swabs yielded negative SARS-CoV-2 test results for all the animals (illustrated in Figure 2 for the lemur that tested seropositive, see below) using an RT-PCR one-tube commercial diagnostic test (Verisure test, from Certest, Zaragoza, Spain). This test, which targets two viral genes (*N* and *ORF1ab*), is intended for humans but it also works with animal samples, as recently proven by ourselves with an otter [13]. Although this test uses the human *RNase P* gene as an internal positive standard, we previously found that this internal control also worked for otters [13] and now we prove this for lemurs (Figure 2). The presence of a positive signal for this internal control (Figure 2) supports the quality of the extracted RNA. We also proved that this assay was working well for the two targeted viral genes by carrying in parallel an external positive control (nasopharyngeal swab RNA from a positive human of our reported series from Sicily [23]), which gave consistently a positive result for these genes in all the assays (Figure 2).

Additional confirmation that the RNA samples from our lemurs did not contain viral RNA was obtained by using a second one-tube commercial SARS-CoV-2 RT-PCR test with these samples, which we had previously shown to work with otters [13]. This test differs from the Viasure test in one of the viral genes targeted (*RdRp* instead of *ORF1ab*) and in the maker (in this case, NZYtech, Lisbon, Portugal). Again, using this test, all lemur samples were negative for amplification of the viral genes, and they were positive for amplification of the *RNaseP*. Furthermore, the external positive control, run in parallel, revealed the amplification of both viral and host genes. 

Since lemur #6 was seropositive for SARS-CoV-2 (see below) but was found negative for the presence of viral RNA by the two commercial one-tube RT-PCR assays, we used in addition with this lemur a two-tube homemade RT-PCR assay believed to be extremely sensitive [4,12], using it in its 3-viral-gene-targeting version (targeted genes, *N*, *ORF10*, and the spike glycoprotein gene *S*) [13,23]. However, this assay also failed to detect the presence of the virus in any of the two occasions in which lemur #6 was sampled (Table 1). Therefore, all the molecular assays led to the conclusion that the seropositive lemur, similarly to all other lemurs of the present study, did not host any SARS-CoV-2 at the dates of any of its two assessments.

### 3.3. Evidence of Infection Based on Serological Findings

As indicated in Materials and Methods, for the determination of normal values in the multispecies ELISA test for anti-RBD IgGs, we first used sera from 17 *V. variegata* and from 21 *L. catta* animals that had been collected prior to the COVID-19 pandemic or in the early pandemic period and that were proven to correspond to animals that had not experienced infection by SARS-CoV-2, as determined by serological criteria. No significant difference was found between the means for the serology results obtained for *V. variegata* and *L. catta*, therefore, the results for both species were merged (total of 38 animals) to estimate a common upper limit of normality for both species, established as an optical density (OD) of 0.16, corresponding to the mean plus 3 standard deviations for the assay results. 

Having defined an upper normality limit for the serological assay, we applied it to all the sera collected from the lemurs of the present study. Among the 59 serological determinations (Figure 3A; all values are means of two determinations differing by less than 10%), only one determination, for the first assessment of lemur #6 (singled out in Figure 3A with black dots), exceeded the upper normality limit of 0.16 OD. For statistical calculations, we excluded this high value as well as the one from the second assessment on this same lemur conducted 8 months later, because the values for this lemur likely reflected previous SARS-CoV-2 infection. The application of the Student’s t-test did not yield a significant difference between the means for both species, as was the case for the initial cohort of sera used for the estimation of the upper limit of normality. Merging the results for both species in our cohort of sera from March 2022 to March 2023, except the sera from lemur #6, a mean ± SD of 0.077 ± 0.023 OD was obtained for the combined results of these 57 sera, which would lead to an upper limit of normality (mean plus three standard deviations) of 0.15 OD, very close to the upper normality limit of 0.16 OD estimated with the initial cohort of non-infected lemurs of both species. In fact, except for lemur #6, none of the sera from the present cohort exceeded a test result of 0.14 (Figure 3A). The negative serology for SARS-CoV-2 in 40 of the 41 lemurs examined between March 2022 and March 2023 (17 of them tested on two occasions separated by 7–11 months) suggests a relatively low susceptibility of these two species of lemurs to SARS-CoV-2 infection, particularly since they were in daily contact with keepers and visitors. This is reassuring concerning policies of interactions of lemurs of these species with humans in humanized habitats or in zoological gardens. 

However, one of these 41 lemurs (lemur #6) was seropositive in one instance and also had a high serological titer, close but below the upper limit of normality, on a second serum sample obtained 8 months after the first extraction (Figure 3A). The seropositive sample was well above the frequency distribution of values of serological results for the rest of the cohort of animals (Figure 3B). Therefore, the picture provided by lemur #6 is consistent with a past infection of this animal with the SARS-CoV-2 virus that gave no clinical notice and that subsided long enough before the first examination of this animal in March 2022. Thus, no virus was detected in the respiratory tract or the intestine and feces, as attested by the negativity of the molecular assays (illustrated in Figure 2 for this animal), but the infection left as a trace an increased blood level of anti-RBD IgG, which was reflected in the high serological titer. A second serum sample collected 8 months later revealed a lower anti-RBD titer, although still at the top of the normality range. This decline in the antibody content fits similar observations in humans following a COVID-19 infection [28]. Therefore, it appears reasonable to conclude that lemur #6 had been infected by the SARS-CoV-2 virus. 

A comparison of the serological results repeated in 18 animals with a 7–11 month interval (Figure 3C) shows that only animal #6 exhibits a significant decrease in the serological titer over time, which, nevertheless, after 7 months, was within the upper range of normality. In most other animals, the titers varied little or even increased somewhat, with only two other instances of decrease (animals #25 and #29), although of much lower absolute magnitude and also much lower values at first determination than with lemur #6. Therefore, it appears that lemur #6 is the only animal that could be experiencing the decrease in post-infection antibodies known to occur in humans following infection [28]. Interestingly, another animal (animal #24) exhibits stable top normal levels similar to the value observed in the second serum extraction of animal #6 (Figure 3C). Actually, the levels in animal #6 would plateau at approximately the same level as in animal #24 if they are assumed to decrease with a t_1/2_ of 1 month (not far from the value for post-infected humans) [28]. In this respect, it is to be noticed in Figure 3B that the range of serological values in the highest part of the normality range (121–140 mOD) encompasses six sera from 5 animals (#6, #17, #24, #26, and #40), following a less populated interval (111–120 mOD), making conceivable earlier SARS-CoV-2 infection leading to stable low level of residual antibodies (known to last long times in humans [28]) in lemurs #17, #24, #26, and #40, in addition to lemur #6.

### 3.4. Final Considerations

To the best of our knowledge, the present study represents the first report of a thorough study of captive lemur populations concerning SARS-CoV-2 infection, conducted on two distinct species of lemurs, ring-tailed lemur (*L. catta*) and black-and-white ruffed lemurs (*V. variegata*), kept in proper care in a highly visited zoological institution. Moreover, this investigation has introduced serological diagnostic methodologies [24,25] into these specific lemur species, filling a void in which there was only one report of serological study in a single wild ruffed lemur as a part of a serological survey of many different types of animals [22].

Our discovery of a seropositive animal among the 23 ruffed lemurs studied here highlights the potential of SARS-CoV-2 to infect at least one (*V. variegata*) of these lemur species and to stimulate an immune response in this animal group. This finding adds to previous knowledge of SARS-CoV-2 transmission to animals, with documented infections of zoo animals but not of lemurs [7,8,9,29]. Such transmission is often considered sporadic, accidental, and largely due to contact with infected keepers [30]. In the present lemur case, the transmission may have occurred through the airborne route, either from an infected worker or from a visiting member of the public, despite the restrictions and hygienic measures applied in the park since 2020; although with the advent of widespread human vaccination, these restrictions were progressively relaxed in 2021 and particularly in 2022 (see Materials and Methods).

Remarkably, this seropositive animal was not noticed to show clinical signs or abnormal findings on physical examination, suggesting that this infection was subclinical. This raises intriguing questions about the overall impact of SARS-CoV-2 on different animal species, particularly when they show few or no perceptible disease manifestations. This scenario mirrors findings in other species, where seropositivity was observed without detectable viral RNA, underscoring the complexity and the potential for subclinical manifestations of SARS-CoV-2 infections in animals [31].

The fact that only one animal was seropositive could raise doubts about the origin of the RBD used in the ELISA, which comes from the ancestral variant (Wuhan), not being the most prevalent variant during the study period. However, the application of this same technique in ferrets during the same study period (with Omicron being the most prevalent) and the detection of several seropositive cases confirm that, in this instance, it did not interfere with the results [24]. Furthermore, it is crucial to emphasize the absence of available information regarding the pattern of anti-SARS-CoV-2 antibodies and cross-reactivity across different SARS-CoV-2 virus variants in the majority of wildlife species. Therefore, judging from the low infection rate of SARS-CoV-2 in our cohort, restricted here to 1–5 animals among a total of 43 animals, the transmission ability of the virus within lemur appears to be limited. In any case, the study of emerging infections in new species of zoo animals is relevant in the context of close and sustained interactions with humans, such as those residing in open facilities with direct contact with the public, a circumstance that is common in many zoological institutions. Additionally, the study of SARS-CoV-2 infections in wild animals is important in the broader context of viral evolution, with the potential for such animals to serve as reservoirs for the virus and to pose a future risk of reinfection in humans. These findings emphasize the need for continued vigilance and research to safeguard both animal and human health in our evolving relationship with SARS-CoV-2.

## 4. Conclusions

In conclusion, our results exclude the presence of active infection in any of the 43 animals including 18 animals in which the molecular assays were performed twice with an interval of 7–11 months. They strongly suggest that one animal (lemur #6) was infected, given its clear seropositivity at the earliest determination and the decay of the titer over 7 months to high values, although within the normality range. They also raise the possibility that four additional animals (#17, #24, #26, and #40) could have been historically infected with SARS-CoV-2, maintaining a memory of that infection through the consistently high, non-declining serological values that set them apart from the remaining 38 animals. These findings indicate a low susceptibility to SARS-CoV-2 in the studied lemur species, suggesting limited virus transmissibility within them. This observation may also suggest the potential efficacy of PPE used by zookeepers, thereby reducing the probability of human-to-lemur transmission. However, they evidence that infection can occur at least in *V. variegata*. In this way, our study sheds much-needed light on the relation of SARS-CoV-2 with captive lemurs held in a highly visited zoo setting, while also developing serological diagnostic techniques for these animals. The identification of a seropositive animal, despite the absence of noticeable clinical symptoms, underscores the potential for accidental and subclinical infections among lemurs kept in captivity in zoological institutions, with epidemiological implications for settings of close interaction with humans. Furthermore, this research emphasizes the need for ongoing surveillance and monitoring of SARS-CoV-2 in animals, as they can act as reservoirs for the virus, potentially posing a future risk of reinfection in humans. This study highlights the complex interplay between human and animal health in the ongoing response to the pandemic.

## Figures and Tables

**Figure 1 animals-14-00140-f001:**
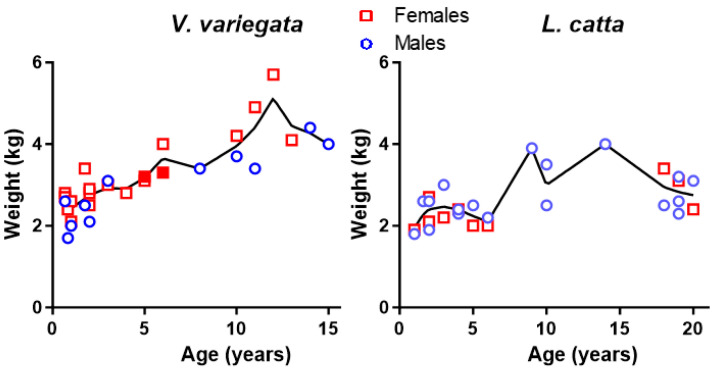
Weights of the lemurs of the two species determined in the present assessments of the lemurs’ health, plotted against the age of the animals on examination. The curves show the Lowess fit (carried out by GraphPad Prism, for 10 points in the smoothing window) for the pooled data for males (blue circles) and females (red squares). In the left panel, the solid red squares correspond to weight determinations in animal #6, the only animal that was seropositive for SARS-CoV-2.

**Figure 2 animals-14-00140-f002:**
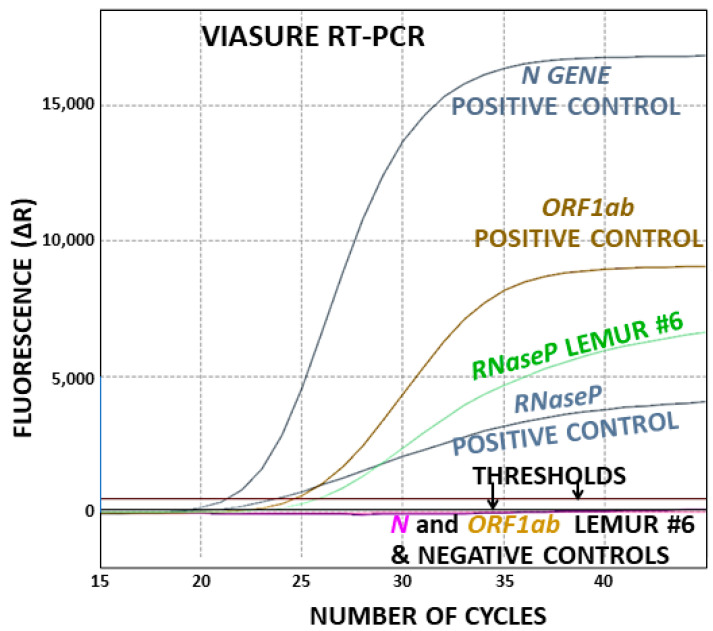
Illustrative example of RT-PCR, for animal 6, the lemur that tested positive by serology, on the date of assessment that gave seropositivity. The sample was RNA from the oropharynx. For details on the procedure and the positive and negative controls, see Section 2.3 Thresholds for positivity were drawn by the program on the basis of the negative controls.

**Figure 3 animals-14-00140-f003:**
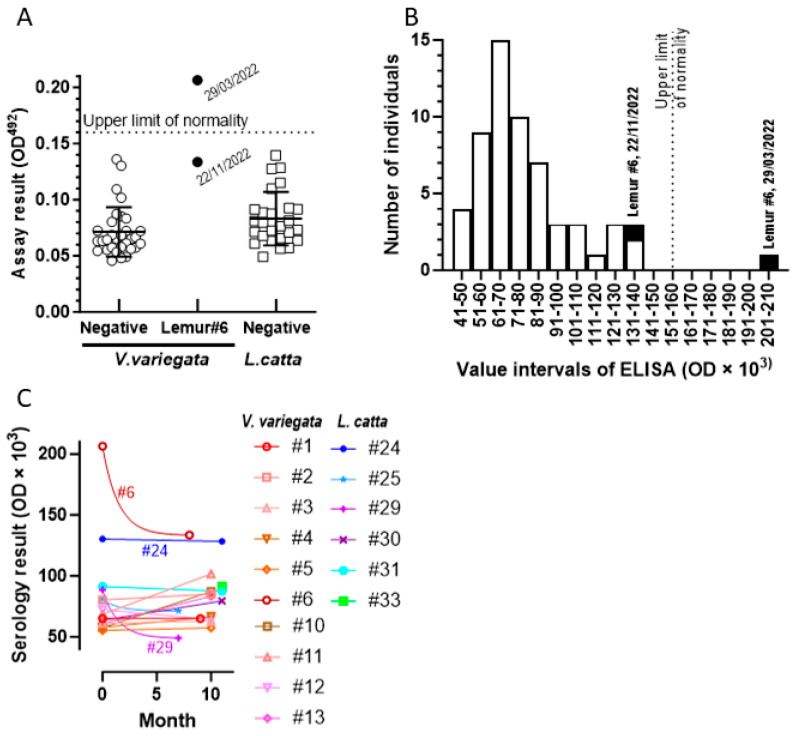
Serology results for anti-RBD of SARS-CoV-2 in the lemurs of the present study. For details, see Materials and Methods. (**A**) Serological titers determined in this study (see Table 1 for the times and animals sampled for serum) grouped per lemur species as indicated. The results for lemur #6 (identified in Table 1; it is a *V. variegata* individual) are singled out in the center as filled circles, indicating the date of the blood extractions. The horizontal lines represent means and standard deviations for the points in the cohort. (**B**) Frequency of serological titers found in our entire animal cohort (pooling of both species) in the indicated intervals of titer values. Results for lemur #6 are in black filling. (**C**) Serology titers for all the animals for which two serum samples were obtained with a 7–11 month interval between the two blood extractions (as indicated). Colors and symbols identify each individual as shown. The two results for each animal are connected with a straight line, except in the three cases, in which the second value was smaller than the first one, in which the lines are the exponentials for a t_1/2_ for the decay of 1 month.

**Table 1 animals-14-00140-t001:** General information about the study animals and their sampling. d/m/y day: Date given as day/month/year. F, M: Female and male, respectively. The symbols + and – mean that the sample was obtained or not obtained, respectively.

No. #	Species	Birth Date (d/m/y)	Sex	Weight (kg)	Sampling Date (d/m/y)	Swab Samples		Serum Sample
						Oropharynx	Rectum	
1	*V. variegata*	03/05/2012	F	4.2	29/03/2022	+	+	+
				4.9	20/12/2022	+	+	+
2	*V. variegata*	22/05/2017	F	3.1	29/03/2022	+	+	+
				4.0	26/01/2023	+	+	+
3	*V. variegata*	24/04/2020	M	2.1	29/03/2022	+	+	+
				3.1	26/01/2023	+	+	+
4	*V. variegata*	05/05/2021	F	2.6	29/03/2022	+	+	+
				3.4	26/01/2023	+	+	+
5	*V. variegata*	13/05/2019	F	3.0	29/03/2022	+	+	+
				2.8	26/01/2023	+	+	+
6	*V. variegata*	24/05/2017	F	3.2	29/03/2022	+	+	+
				3.3	22/11/2022	+	+	+
7	*V. variegata*	21/05/2009	F	4.1	29/03/2022	+	+	+
8	*V. variegata*	24/04/2020	F	2.8	29/03/2022	+	+	+
9	*V. variegata*	08/05/2010	F	5.7	29/03/2022	+	+	+
10	*V. variegata*	09/06/2008	M	4.4	29/03/2022	+	+	+
				4.0	25/01/2023	+	+	+
11	*V. variegata*	No data	M	3.9	29/03/2022	+	+	+
				4.0	25/01/2023	+	+	+
12	*V. variegata*	11/05/2012	M	3.7	29/03/2022	+	+	+
				3.4	26/01/2023	+	+	+
13	*V. variegata*	02/05/2022	M	2.0	02/05/2022	+	+	+
				2.5	26/01/2023	+	+	+
14	*V. variegata*	02/05/2022	F	2.8	26/01/2023	+	+	+
15	*V. variegata*	01/05/2022	F	2.7	22/11/2022	+	+	+
16	*V. variegata*	07/05/2014	M	No data	29/03/2022	+	+	+
17	*V. variegata*	05/05/2021	M	1.7	29/03/2022	+	+	+
18	*V. variegata*	24/04/2020	F	2.5	29/03/2022	+	+	+
19	*V. variegata*	07/05/2014	M	3.4	29/03/2022	+	+	+
20	*V. variegata*	05/05/2021	F	2.1	29/03/2022	+	+	+
21	*V. variegata*	24/04/2020	F	2.9	29/03/2022	+	+	+
22	*V. variegata*	01/05/2021	F	2.4	29/03/2022	+	+	+
23	*V. variegata*	01/05/2022	M	2.6	26/01/2023	+	+	-
24	*L. catta*	26/03/2004	F	3.4	29/03/2022	+	+	+
				3.1	02/03/2023	+	+	+
25	*L. catta*	21/03/2021	M	1.8	29/03/2022	+	+	+
				2.6	19/10/2022	+	+	+
26	*L. catta*	23/08/2012	M	2.5	29/03/2022	+	+	+
27	*L. catta*	27/03/2013	M	3.9	29/03/2022	+	+	+
28	*L. catta*	02/08/2019	F	2.4	02/03/2023	+	+	+
29	*L. catta*	18/04/2020	M	2.6	29/03/2022	+	+	+
				3.0	19/10/2022	+	+	+
30	*L. catta*	10/03/2017	F	2.0	29/03/2022	+	+	+
				2.0	02/03/2023	+	+	+
31	*L. catta*	17/03/2021	F	1.9	29/03/2022	+	+	+
				2.7	02/03/2023	+	+	+
32	*L. catta*	31/03/2008	M	4.0	29/03/2022	+	+	+
33	*L. catta*	11/03/2020	F	2.1	29/03/2022	+	+	+
				2.2	02/03/2023	+	+	+
34	*L. catta*	15/03/2016	M	2.2	12/04/2022	+	+	+
35	*L. catta*	21/03/2003	M	2.3	12/04/2022	+	+	+
36	*L. catta*	26/03/2004	M	2.5	12/04/2022	+	+	+
				3.2	02/11/2022	+	+	+
37	*L. catta*	08/03/2003	M	2.6	12/04/2022	+	+	+
				3.1	02/11/2022	+	+	+
38	*L. catta*	13/03/2020	M	1.9	06/04/2022	-	+	+
39	*L. catta*	26/06/2018	M	2.3	06/04/2022	-	+	+
40	*L. catta*	11/03/2017	M	2.5	06/04/2022	+	+	+
41	*L. catta*	18/06/2018	M	2.4	06/04/2022	+	+	+
42	*L. catta*	12/05/2013	M	3.5	10/01/2023	+	+	-
43	*L. catta*	27/03/2003	F	2.4	02/03/2023	+	+	+

## Data Availability

The data used to support the findings of this study are included within the article.

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
