# Peer review of "Molecular and Serological Studies on Potential SARS-CoV-2 Infection among 43 Lemurs under Human Care—Evidence for Past Infection in at Least One Individual"

_animals, 2023, doi:10.3390/ani14010140_

Round 1

Reviewer 1 Report

Comments and Suggestions for Authors

Dear Authors, it is an very interesting manuscript because specielly in a park the virus transmission is mportant for the health status of all animals and for human beings. Please correct the marked sentences.

1. The main question addressed by the research is SARS CoV-2 in wild animals.
2. In zoos it is very important that wlld animals can have SASRS CoV-2 and spread to human beings.
3. The wild animal lemur can have SARS CoV-2.
4. The PCR test is important to test in wild animals too.
5. The conclusions are consistent with the evidence and arguments presented and they address the main question posed.
6. The references are appropriate.
7. Tables are correct.

Author Response

REPLIES TO REVIEWER 1

We thank Reviewer 1 for her/his highly constructive comments. Detailed responses follow, with details of the changes made.

Query 1. please correct: 23 L. catta, 20 V. variegata ...

Answer: Done. Now reads as follows” 23 and 20 of them belonging, respectively, to the V. variegata and L. catta species” (Lines 115 in the clean version)

Query 2. Please include: From what vene was blood collected?

Answer: The vein has been added in the revised manuscript and now reads as follows “and collection of venous blood samples from the jugular vein for hematocrit,” (Lines 154 in the clean version).

Reviewer 2 Report

Comments and Suggestions for Authors

This strikes me as a carefully conducted study, clearly presented in this manuscript, which provides useful information on the transmission of the SARS-CoV-2 virus from humans to two species of lemur during the course of limited interaction in a zoo setting.  Because, as it reports, significant measures were taken to limit contact between humans and the lemurs during 2022, the period of study, including use of PPE for all zookeepers, the humans who had closest contact with the lemurs, the very limited numbers of lemurs testing positive for COVID-19 could have been a result of either that limitation of contact or poor transmission of the virus from humans to lemurs. The study’s findings are of interest, but its conclusion should make clear that both of these underlying dynamics were at play, and it cannot be concluded that there is poor transmission of the virus from humans to lemurs.

The background on other studies of transmission of COVID-19 between humans and animals is well rounded. The authors may also wish to cite the following additional study, which reports a comprehensive summary of such transmission: Meekins, D.A.; Gaudreault, N.N.; Richt, J.A. Natural and Experimental SARS-CoV-2 Infection in Domestic and Wild Animals. Viruses 2021, 13, 1993.

Tables 1-3 present data that it seems could be relegated to supplementary material, with summaries of each included as the corresponding tables shown in the text.

One minor editorial note: in line 197, the use of either a comma or period for a decimal separator should be consistent (“prevalence of SARS-CoV-2 infection of 2.32 % in both species and 4,34% in V. variegate”).

With these notes, this manuscript presents data of interest regarding transmission of COVID-19 between humans and lemurs, and I recommend publication with consideration of the points above.

Author Response

REPLIES TO REVIEWER 2

We thank Reviewer 2 for her/his highly constructive comments. Detailed responses follow, with details of the changes made.

Query 1: This strikes me as a carefully conducted study, clearly presented in this manuscript, which provides useful information on the transmission of the SARS-CoV-2 virus from humans to two species of lemur during the course of limited interaction in a zoo setting.  Because, as it reports, significant measures were taken to limit contact between humans and the lemurs during 2022, the period of study, including use of PPE for all zookeepers, the humans who had closest contact with the lemurs, the very limited numbers of lemurs testing positive for COVID-19 could have been a result of either that limitation of contact or poor transmission of the virus from humans to lemurs. The study’s findings are of interest, but its conclusion should make clear that both of these underlying dynamics were at play, and it cannot be concluded that there is poor transmission of the virus from humans to lemurs.

Answer 1: Thank you for your valuable insights during the review process. We have taken your suggestion seriously and incorporated it into the conclusion of our manuscript. Specifically, we have highlighted the possibility of effective Personal Protective Equipment (PPE) utilization by zookeepers as a potential contributing factor to the low incidence of SARS-CoV-2 among lemurs. Now reads as follows: “These findings indicate a low susceptibility to SARS-CoV-2 in the studied lemur species, suggesting limited virus transmissibility within them. This observation may also suggest the potential efficacy of PPE used by zookeepers, thereby reducing the probability of human-to-lemur transmission. In addition, the findings could be attributed to a combination of both factors.”  (line 460-464 in the clean version)

Query 2: The background on other studies of transmission of COVID-19 between humans and animals is well rounded. The authors may also wish to cite the following additional study, which reports a comprehensive summary of such transmission: Meekins, D.A.; Gaudreault, N.N.; Richt, J.A. Natural and Experimental SARS-CoV-2 Infection in Domestic and Wild Animals. Viruses 2021, 13, 1993.

Answer: Thank you for your valuable suggestion. We have taken it into consideration and incorporated the reference you provided into the discussion section of the manuscript. In the revised manuscript is the reference 30 and now reads as follows: “This finding adds up to previous knowledge of SARS-CoV-2 transmission to animals, with documented infections of zoo animals but not of lemurs [7-9, 30].”(line 424-426 in the clean version)

Query 3: Tables 1-3 present data that it seems could be relegated to supplementary material, with summaries of each included as the corresponding tables shown in the text.

Answer: Thank you for your insightful feedback regarding the tables in our manuscript. We have taken your suggestions into account and have made significant improvements. Specifically, Tables 1 and 2 have been merged for better coherence and readability. Additionally, Table 3 has been transformed into Figure 3, enhancing its visual representation and explanatory value. We believe these changes have significantly improved the presentation of our data.

Query 4: One minor editorial note: in line 197, the use of either a comma or period for a decimal separator should be consistent (“prevalence of SARS-CoV-2 infection of 2.32 % in both species and 4,34% in V. variegate”).

Answer: We have reviewed and revised the document accordingly. We have decided to maintain the use of points as decimal separators throughout the entire manuscript for consistency.

Query 5: With these notes, this manuscript presents data of interest regarding transmission of COVID-19 between humans and lemurs, and I recommend publication with consideration of the points above

Answer: Thank you for your thorough review and recommendations. We appreciate your acknowledgment of the manuscript's contribution regarding the transmission of COVID-19 between humans and lemurs. We've taken your notes into careful consideration, and based on your insights, we believe the data presented holds significant interest.

Reviewer 3 Report

Comments and Suggestions for Authors

The present work analyses the circulation of SARS-CoV-2 among two population of captive lemurs in a zoological park in Spain, focusing on two different sampling periods before and after the re-opening of the park to the public after limitations due to the pandemics.

The idea of this work is noteworthy: monitoring the circulation of this virus in animal species that have a close contact with humans is definetly of interest, especially amon non-human primates.

Nonetheless, there are:

1) some methodological limitations that need to be explained and discussed due to their relevance for the conclusions;

2) some changes in the presentations of the results that would facilitate the reading of the present manuscript.

1) While the molecular analysis is sound and well-described, the serological one has one main issue: it does not take into account viral mutations and the emergece on novel variants, neither in the methods nor in the discussion.

The serological analysis is based on an ELISA test screening for anti-RBD antibodies, but the authors never explain to which variant this RBD belongs. The references 26 and 27, cited in the materials and methods sections, have been published in early 2021, so are probably based on the original strain from Wuhan originated in 2020. However, this strain is no longer circulating and was not circulating in Spain at the time of sampling, so that a negative result might not really exclude the prior infection of the animal with a more recent variant. This should be clarified in the materials and methods section 
(which variant was tested) and considered during the discussion.

2) The manuscript has been submitted as a Brief Report, but it is 17 pages long due to the presence of 3 redundant tables. There is no need for a double identification of the individuals (ID and internal ID). The sex, age and weight of the animals are not relevant for this study and could be omitted ore moved to the supplementary materials. The samples collected are the same for every individual, with just a few exceptions that can be easily described in the main text or in a caption. The remaining data can fit in just one table.

Other line-by-line suggestions:

line 24: "This study studies" to become "This study evaluates"

l. 58: "specific domesticated animals" to become "some domesticated animals"

l. 112: would add reference to the IUCN website

Table 1: in addition to what I said before, date format is not the same for all the dates, and the Genera can be abbreviated to optimize space while adding new columns from table 3.

l. 147: section title should be on the left

l. 144-170: I think you should anticipate at the beginning of this section the samples that were generally collected, and later explain the exceptions where serum or swabs were not taken. By doing this, table 2 can be easily removed.

l. 220: as I said, the tested variant(s) should be detailed.

l. 250: "For the cut-off for lemur", I think the first "For" should be removed

l. 252-255: "performed" should become "assessed". In any case, I did not understand this sentence, and where does this anti-N Elisa come from. To my understanding, testing the negative the animals and comparing with sera collected before the pandemics is enough. Instead, if you tested the samples with different ELISA tests and a SNT, these should be well described in the methods and not just mentioned.

Table 3: caption should be fixed

Discussion: as said before.

Refs 8 and 20 need to be fixed: in the first the authors are missing, in the latter they are in capitals.

Comments on the Quality of English Language

English Language is pretty fine, but there are a couple of minor issues that are probably due to manuscript revisions before submitting.

Author Response

REPLIES TO REVIEWER 3:

We thank Reviewer 3 for her/his highly constructive comments. Detailed responses follow, with details of the changes made.

Query 1: The present work analyses the circulation of SARS-CoV-2 among two population of captive lemurs in a zoological park in Spain, focusing on two different sampling periods before and after the re-opening of the park to the public after limitations due to the pandemics.The idea of this work is noteworthy: monitoring the circulation of this virus in animal species that have a close contact with humans is definetly of interest, especially amon non-human primates.

Answer: Thank you for acknowledging the focus of our study, which examines SARS-CoV-2 circulation among captive lemurs in a Spanish zoological park pre and post its reopening during pandemic limitations. We aimed to monitor virus circulation in species closely interacting with humans, especially non-human primates. Your recognition of the significance of this approach is greatly appreciated.

Query 2: some methodological limitations that need to be explained and discussed due to their relevance for the conclusions;

Answer: Thank you for highlighting the importance of methodological clarity. We've taken your suggestion seriously and have revised the molecular and serological sections to provide comprehensive explanations. These revisions aim to address the methodological limitations more explicitly, ensuring a thorough discussion that directly influences our conclusions.

Query 3: some changes in the presentations of the results that would facilitate the reading of the present manuscript.

Answer: Thank you for your valuable feedback. In response to your suggestion, we've combined the results and discussion sections to enhance the coherence and readability of the manuscript. Additionally, we've incorporated three figures summarizing the results, aiming to provide a more visual and comprehensive understanding. In addition, we have fused table 1 and 2 in only one table. These changes aim to improve the clarity and accessibility of our findings for the reader's convenience. Now Table 1, Figure 1, 2 and 3 can be found.

Query 4: 1) While the molecular analysis is sound and well-described, the serological one has one main issue: it does not take into account viral mutations and the emergece on novel variants, neither in the methods nor in the discussion.The serological analysis is based on an ELISA test screening for anti-RBD antibodies, but the authors never explain to which variant this RBD belongs. The references 26 and 27, cited in the materials and methods sections, have been published in early 2021, so are probably based on the original strain from Wuhan originated in 2020. However, this strain is no longer circulating and was not circulating in Spain at the time of sampling, so that a negative result might not really exclude the prior infection of the animal with a more recent variant. This should be clarified in the materials and methods section (which variant was tested) and considered during the discussion.

Answer: Antibodies to SARS-CoV-2 were determined by an indirect ELISA for the detection of IgG specific for RBD (Ancestral SARS-CoV-2 strain, Wuhan strain). In our case, we did not have different variants of the virus in our laboratory. Therefore, we used those we had available the Wuhan variant for ELISA. We have performed this ELISA composed by Wuhan strain to detect the presence of anti-SARS-CoV-2 antibodies in ferrets from January 2022 to May 2023 being the prevalent SARS-CoV-2 strain Omicron (B.1.1.529) Sublineages including BA.1, BA.2, BA.5, BQ.1, BQ.1.1. In this study, seropositive ferrets were detected during this period of time (Giner et al., 2023). However, it is important to note that there is no available information related to the pattern of anti-SARS-CoV-2 antibodies and cross-reactivity based on different SARS-CoV-2 virus variants in most wildlife animals.  Therefore we have included the following sentences in :

  • Methodology: “An indirect enzyme-linked immunosorbent assay (ELISA) that detects IgG recognizing the receptor-binding domain (RBD) of the spike protein (S) of SARS-CoV-2 virus (Ancestral SARS-CoV-2 strain, Wuhan strain) in multiple animal species was used as previously described [25,26]. Unfortunately, despite the prevailing presence of the SARS-CoV-2 Omicron variant (B.1.1.529) during our period study, this specific variant was not accessible in our laboratory for analysis. However, this limitation did not hin-der the detection of seropositive ferrets using the same technique within the study pe-riod in a previous work [25].Unfortunately, despite the prevailing presence of the SARS-CoV-2 Omicron variant (B.1.1.529) during our period study, this specific variant was not accessible in our la-boratory for analysis. However, this limitation did not hinder the detection of sero-positive ferrets using the same technique within the study period in a previous work [25].” (lines 221-228 in the clean version)
  • Results and Discussion: “The fact that only one animal is seropositive could raise doubts about the origin of the RBD used in the ELISA, which comes from the ancestral variant (Wuhan), not being the most prevalent variant during the study period. However, the application of this same technique in ferrets during the same study period (with Omicron being the most prevalent) and the detection of several seropositive cases confirm that, in this instance, it did not interfere with the results [25].” (lines 445-450 of the clean version)
  • References: We have replaced reference 25 by a reference that was detected seropositives in ferrets in the same study period with the same technique "Giner, J., Lebrero, M. E., Trotta, M., Rueda, P., Vilalta, L., Verde, M., Hurtado-Guerrero, R., Pardo, J., Lacasta, D., Santiago, L., Arias, M., Peña-Fresneda, N., Montesinos, A., Pérez, M. D., Fernández, A., & Villanueva-Saz, S. (2023). Seroprevalence of anti-SARS-CoV-2 antibodies in household domestic ferrets (Mustela putorius furo) in Spain, 2019-2023. Veterinary research communications, 10.1007/s11259-023-10190-2. Advance online publication. https://doi.org/10.1007/s11259-023-10190-2: “

Query 5: The manuscript has been submitted as a Brief Report, but it is 17 pages long due to the presence of 3 redundant tables. There is no need for a double identification of the individuals (ID and internal ID). The sex, age and weight of the animals are not relevant for this study and could be omitted ore moved to the supplementary materials. The samples collected are the same for every individual, with just a few exceptions that can be easily described in the main text or in a caption. The remaining data can fit in just one table.

 Answer: Thank you for your insightful suggestions. We've addressed all of them in the revised manuscript. We've condensed the redundant information from tables 1 and 2 into a single table (Table 1) and transformed the previous Table 3 into a more illustrative and concise Figure 3. Additionally, following your recommendation, we have removed the double identification of individuals (ID and internal ID). The sample collection information has been succinctly summarized for clarity within the main text or appropriately captioned. These modifications have significantly reduced the length of the manuscript while retaining all relevant data in a more concise and cohesive format.

Minor comments:

MC1: line 24: "This study studies" to become "This study evaluates"

Answer: Thank you for your feedback. We've taken extensive measures to enhance the manuscript's overall grammar and language quality. As a result of these revisions, the sentence in question has been removed from the text. We believe these improvements have significantly elevated the clarity and readability of the manuscript.

MC2: l. 58: "specific domesticated animals" to become "some domesticated animals"

Answer: Done. Now reads as follows: “infection in some domestic and farmed animals” (line 61-62 of the clean version)

MC3:l. 112: would add reference to the IUCN website

Answer: Done. “two species in the IUCN Red List of Threatened Species as endangered and critically endangered, respectively; https://www.iucnredlist.org)” (line 112-144 of the clean version)

MC4:Table 1: in addition to what I said before, date format is not the same for all the dates, and the Genera can be abbreviated to optimize space while adding new columns from table 3.

Answer: Table 1 and 2 were consolidated and streamlined, while Table 3 was substituted with Figure 3.

MC5:l. 147: section title should be on the left

Answer: Done (line 152 of the clean version)

MC6: l. 144-170: I think you should anticipate at the beginning of this section the samples that were generally collected, and later explain the exceptions where serum or swabs were not taken. By doing this, table 2 can be easily removed.

Answer: We have fused table 1 and 2 (Table 1) and we have explained better in that section.

MC7:l. 220: as I said, the tested variant(s) should be detailed.

Answer: Thank you for your comment. We have addressed this concern in our response to Query 4, specifically in lines 221-224 in the clean version, where we now provide detailed information about the tested variant as follows  “ An indirect enzyme-linked immunosorbent assay (ELISA) that detects IgG recognizing the receptor-binding domain (RBD) of the spike protein (S) of SARS-CoV-2 virus (An-cestral SARS-CoV-2 strain, Wuhan strain) in multiple animal species was used as pre-viously described [25,26].  “

MC8: l. 250: "For the cut-off for lemur", I think the first "For" should be removed

Answer: Done. Now reads as follows “The cut-off for our present indirect ELISA assay for anti-RBD IgG for lemurs…” (line 241 of the clean version)

MC9:l. 252-255: "performed" should become "assessed". In any case, I did not understand this sentence, and where does this anti-N Elisa come from. To my understanding, testing the negative the animals and comparing with sera collected before the pandemics is enough. Instead, if you tested the samples with different ELISA tests and a SNT, these should be well described in the methods and not just mentioned.

Answer: Thank you for your comments and suggestions regarding the methodology section. The methodology in question, involving the assessment of samples using various ELISA tests and an anti-N ELISA, has been removed from the manuscript. This was due to the confirmation of negative results obtained from another laboratory, leading us to prefer a clearer presentation without potential confusion for the readers. I hope this finds well.

MC10:Table 3: caption should be fixed

Answer: Table 3 has been replaced by Figure 3.

MC11:Discussion: as said before.

Answer: Thank you for your comment. We have addressed this concern in our response to Query 4, specifically in lines 445-450, and now reads as follows “The fact that only one animal is seropositive could raise doubts about the origin of the RBD used in the ELISA, which comes from the ancestral variant (Wuhan), not being the most prevalent variant during the study period. However, the application of this same technique in ferrets during the same study period (with Omicron being the most prevalent) and the detection of several seropositive cases confirm that, in this instance, it did not interfere with the results [25].”

MC12:Refs 8 and 20 need to be fixed: in the first the authors are missing, in the latter they are in capitals.

Answer: Thank you for highlighting the issues with references 8 and 20 (now references 7 and 19). We have thoroughly reviewed the references and made the necessary corrections. The missing authors in reference 8 (in the revised version reference 7) have been added, and in reference 20 (in the revised version 19) , the authors' names are now formatted correctly without capitalization. We appreciate your attention to detail and ensuring the accuracy of our reference list.

Round 2

Reviewer 3 Report

Comments and Suggestions for Authors

Dear authors,

I would like to thank you for taking this revision activity so seriously. I strongly believe that you have understood that my previous queries were meant to improve the quality of this paper: with these new figures, a modified table and, most importantly, an extensive revision of the results and discussion, it is now much more easy and interesting to read.

I understand that no serological analysis for other variants was possible for this work, now it is clear and well discussed. Anyway, I strongly recommend to face this issue for future projects.

I'm happy to endorse the publication of your work, with just the following minor edits:

1) lines 99-108: I would remove this final paragraph from the introduction: it is a discussion of the work, that is well done later in the text and has no need to be anticipated here (it's already mentioned in the abstract as well).

2) line 239: "period study" should be "study period"

3) line 277 and 333: fix "V. variegate"

4) I guess it's due to the track-changes mode, but the caption of fig.3 is above the figure instead of being below. I would suggest, if possible, to enlarge the graph of fig. 3C, to make it clearer to appreciate: maybe the legend could be shrinked?

5) line 478: comma to be moved after "values"

Author Response

REPLIES TO REVIEWER 3:

We thank Reviewer 3 for her/his highly constructive comments. Detailed responses follow, with details of the changes made.

Query 1: I would like to thank you for taking this revision activity so seriously. I strongly believe that you have understood that my previous queries were meant to improve the quality of this paper: with these new figures, a modified table and, most importantly, an extensive revision of the results and discussion, it is now much more easy and interesting to read. I understand that no serological analysis for other variants was possible for this work, now it is clear and well discussed. Anyway, I strongly recommend to face this issue for future projects.

Answer: Thank you for acknowledging the improvements made in response to your feedback. The inclusion of new figures, a revised table, and a comprehensive rework of the results and discussion aimed to enhance readability, as noted. We appreciate your understanding regarding the limitations of serological analysis for other variants in this study and will consider this for future projects. Your guidance has been invaluable in refining our work.

Minor comments:

MC1: lines 99-108: I would remove this final paragraph from the introduction: it is a discussion of the work, that is well done later in the text and has no need to be anticipated here (it's already mentioned in the abstract as well).

Answer: Thank you for your insightful suggestion regarding the final paragraph in the introduction (lines 99-108 in the revised version). We have duly considered your recommendation and removed the paragraph as per your suggestion.

MC2: line 239: "period study" should be "study period"

Answer: Done. Now reads as follows: “Unfortunately, despite the prevailing presence of the SARS-CoV-2 Omicron variant (B.1.1.529) during our study period” (line 225 of the revised version)

MC3: line 277 and 333: fix "V. variegate"

Answer: Done. We have revised the whole manuscript and we have changed the two timed that were wrong written (lines 263 and 319 of the revised version)

MC4: I guess it's due to the track-changes mode, but the caption of fig.3 is above the figure instead of being below. I would suggest, if possible, to enlarge the graph of fig. 3C, to make it clearer to appreciate: maybe the legend could be shrinked?

Answer: Thank you for your feedback. We've addressed the issue with the figure caption placement, ensuring it now appears below Figure 3 as intended. Regarding your suggestion to enlarge the graph of Figure 3C for improved clarity, we've taken steps to enhance its visibility while considering shrinking the legend to accommodate this change.

MC5: line 478: comma to be moved after "values"

Answer: Done (line 462 of the revised version)
